# RESOURCE EFFICIENT SELF-SUPERVISED LEARNING FOR SPEECH EMBEDDINGS

## ABSTRACT

Representation learning from sequential data using self-supervised learning (SSL) has proven to be a powerful technique and improved state-of-the-art (SOTA) results when fine-tuned for various downstream tasks. So far the success of SSL frameworks, e.g., Wav2Vec2 and Data2Vec2, for learning audio embeddings is primarily carried out by masking intermediate features and then solving a contrastive or non-contrastive task in an end-to-end manner, respectively. In comparison to contrastive SSL methods such as Wav2Vec2, non-contrastive techniques such as Data2Vec2 have emerged having better model quality and training time. However, Data2Vec2 is still quite demanding in terms of resources, namely infrastructure (more and better GPUs), which remains a significant barrier to further improving models for downstream tasks. In this work we show that non-contrastive learning, such as an extension of the Barlow–Twins methodology, when applied to a range of downstream tasks simultaneously decreases training time and resource requirements while maintaining or improving SOTA results in key benchmark datasets. From a computional point of view, our approach permits effective training with smaller sequence lengths and batch sizes without requiring gradient accumulation, compared to Data2Vec2, reducing GPU VRAM requirements from NVIDIA A100's to V100's.

## 1 INTRODUCTION

Recent progress in self-supervised learning (SSL) has been highly successful in utilizing unlabeled data and demonstrated superior performance in the domains of computer vision (CV) Chen et al. (2020); He et al. (2020); Chen & He (2021), natural language processing (NLP) Devlin et al. (2019); Lewis et al. (2019), and speech recognition (SR) Liu et al. (2020); Chung et al. (2021); Baevski et al. (2022b); Schneider et al. (2019); Baevski et al. (2020). In particular, SSL-based approaches exploit abundance of unlabeled data to learn underlying representations, while using both *contrastive* and *non-contrastive* approaches Jaiswal et al. (2020); Balestriero & LeCun (2022). Especially, in the domain of ASR, masking based contrastive and non-contrastive methods have emerged as the leading SSL approaches and yielding current state-of-the-art (SOTA) solutions, e.g., Wav2Vec2 Baevski et al. (2020) and HuBERT Hsu et al. (2021) for contrastive approaches and Data2Vec Baevski et al. (2022b) and Data2Vec2 Baevski et al. (2022a) for non-contrastive methods. The success of these approaches is mainly due to easy availability of large curated unlabeled open source datasets Kearns (2014); Panayotov et al. (2015); Kahn et al. (2020); Wang et al. (2021); Ardila et al. (2020), availability of industry-scale GPU infrastructures, improvements in the data training pipeline and scaling (e.g., data-, pipeline-, model-parallelism) of deep learning frameworks. Prior studies Baevski et al. (2022a) have shown that non-contrastive approaches yield lower training time and require fewer computational resources compared with contrastive methods, but nevertheless the overall training time and resource requirements for achieving SOTA performance remains a significant barrier to further improving ASR solutions using SSL.

In this paper, we propose a new non-contrastive method for learning speech representations, which reduces training time and decreases resource requirements while improving SOTA results in key LibriSpeech benchmarks datasets. We do so by breaking away from previous non-contrastive SSL for ASR which is mainly based on masking intermediate features, and instead develop a non-masking approach. Specifically, we consider Barlow–Twins Zbontar et al. (2021) (BT) as a representative example of non-contrastive SSL in the image domain and expand its scope from vision to audio

by inventing the following extensions: (i) we incorporate a number of new loss functions via purposefully designed time-merging and time-unrolling methods, and (ii) applying static (hyper-parameter optimization) and dynamic (stop gradient) methodologies to balance the different scales in individual losses.

Furthermore, rather than being limited by a dichotomy between constrastive and non-contrastive methods, we explore the effect of the sequential use of contrastive and non-contrastive training and observe improved performance, i.e., decreased word error rate (WER) when compared to solely contrastive or non-contrastive training, which is in line with recent work on SSL based speech representation learning for speaker verification rather than ASR Zhang & Yu (2022).

A summary of our main findings regarding the benefits of our approach for speech representation learning are as follows: (i) our proposed non-contrastive BT SSL ASR yields SOTA results in key benchmark datasets, (ii), our new non-contrastive BT method requires fewer GPUs and smaller batch sizes thereby reducing memory requirements compared with non-contrastive Data2Vec2 methods, which allows cheaper infrastructure investment from NVIDIA A100's to V100's, and (iii) lowest WER can be achieved by sequentially combining contrastive Wav2Vec2 approach followed with our proposed non-contrastive BT training.

## 2 APPROACH

The most common SSL methods in speech considered are masking-based contrastive learning and autoregressive prediction based learning. In this work, we explore the potential of a non-contrastive SSL method for learning speech representations and its effectiveness on the downstream ASR.

### 2.1 MOTIVATION

Recent work in the area of non-contrastive SSL (e.g., BYOL (Grill et al., 2020), SimSiam (Chen & He, 2021), Barlow–Twins (Zbontar et al., 2021), DINO (Caron et al., 2021)) have shown remarkable capacity to learn powerful representations from only positive pairs, i.e., two augmented views of the same data point. Unlike contrastive SSL approaches that use negative pairs to prevent representational collapse, non-contrastive SSL approaches employ a dual pair of Siamese networks to process two augmented views of a data point and minimize their representational differences.

In general, contrastive SSL methods require large batch sizes, e.g., SimCLR (Chen et al., 2020) and MoCo (He et al., 2020), to achieve good performance. On the contrary, non-contrastive SSL approaches are comparatively more efficient and easy to train with smaller batches and reduced memory. As shown in (Zbontar et al., 2021), a non-contrastive SSL method such as Barlow–Twins could learn effectively with up to 16x smaller batch size.

### 2.2 METHOD

In this subsection, we present an overview of our approach for learning speech embeddings with a first of its kind non-contrastive SSL method designed for time series speech modeling (c.f., Figure 1 (b)) and its comparison with a standard non-contrastive SSL method for non-time series data such as images (see Figure 1 (a)).

Similar to all non-contrastive SSL methods used in vision, our approach for learning speech embeddings has a dual pair of Siamese networks referred to as online ($O$) and target ($T$) networks. Only the online network is trained via gradient descent and the target network employs a momentum encoder (He et al., 2020) that slowly follows the online network in a delayed fashion through an exponential moving average (EMA). The outputs of the online and target networks are then encouraged to learn good representations via a self-supervised loss function.

However, there are two key differences in our approach for learning speech embeddings compared to image embeddings, which can be categorized as modeling and learning differences. These differences are summarized below.

First, instead of performing augmentations in the input space (c.f. Figure 1 (a)), our solution operates in a latent space (see Figure 1 (b)). Specifically, we apply augmentation not directly on the input $X$, but rather to the outputs of the feature extractor to generate $Y^A$ and $Y^B$. The motivation behind this is that the most common form of data augmentation in audio is SpecAugment (Park et al., 2019) which acts on the audio spectogram. However in the Wav2Vec2 (Baevski et al., 2020)

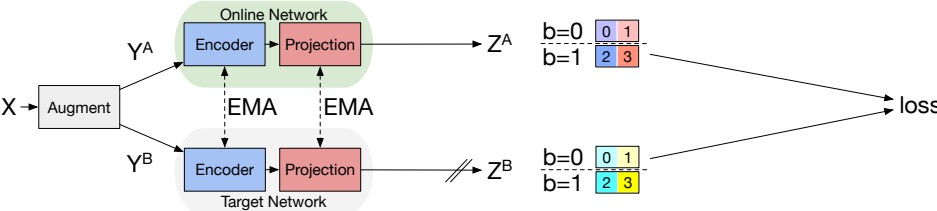

(a) Standard non-contrastive SSL approach with optional exponential moving average (EMA) and stop gradient (sg) to learn representation of non-time series data. Such an approach takes image data $X$ (depicted with batch size b=2), augments it into two distorted views $Y^A$ and $Y^B$, and feeds it via online and target networks to produce representational embeddings $Z^A$ and $Z^B$. An appropriate loss function is then created from these embeddings for self-supervised learning; particularly relevant to our work, the existing Barlow–Twins loss function cross-correlates $Z^A$ and $Z^B$ and promotes the correlation matrix to be close to identity.

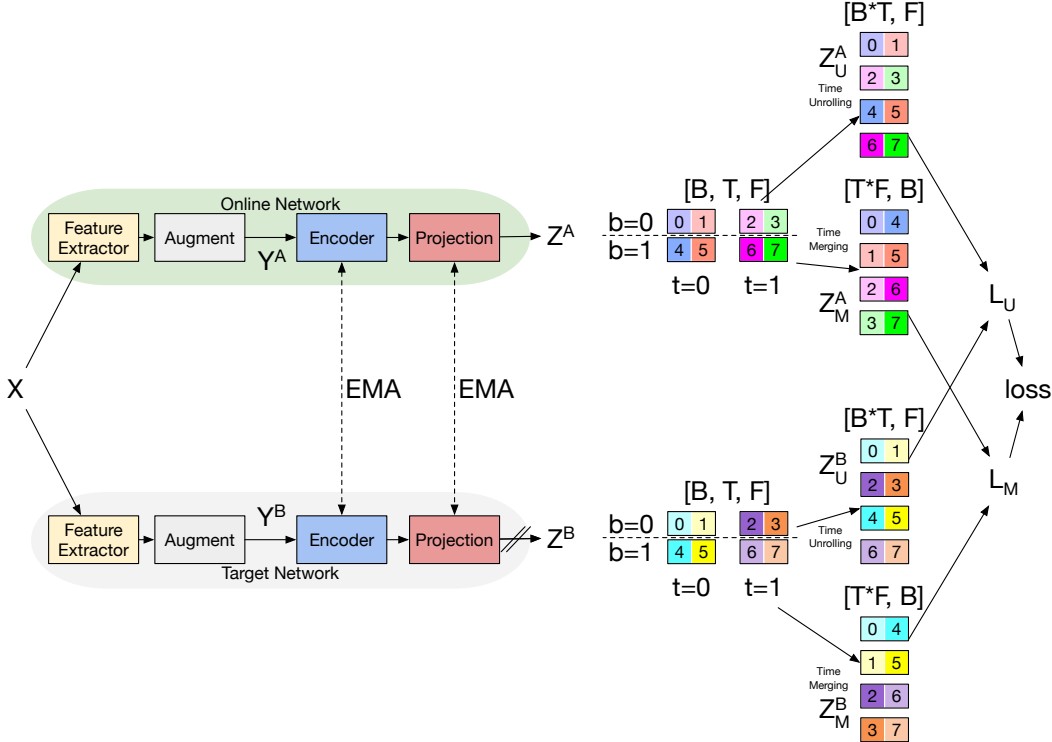

(b) Our approach for learning time series speech representations via non-contrastive SSL method. In the audio domain, a larger encoder is often composed of two submodules namely a feature extractor and a smaller encoder as depicted in the figure, and data augmentation employed in between. This motivates the changes to the online and target networks in our approach. Furthermore, traditional loss functions such as Barlow–Twins cannot be readily used given the dimensionality of the embeddings $Z$ is larger in the audio than the image domain. For this reason we introduce time unrolling and merging approaches leading each to a different loss function which we statically or dynamically combine for improved training.

Figure 1: Comparison of non-contrastive SSL approaches for (a) image and (b) audio data.

architecture the spectogram is replaced by a learnable sub-network referred to as the *feature extractor*. Nevertheless we have found that the same principles underlying SpecAugment are used at this point in the processing pipeline in order to avoid overfitting and increase generalization.

The second key difference stems from the way the loss is computed from the produced embeddings $Z^A$ and $Z^B$. Depending on if the scenario is non-time series or time series there is a fundamental difference between the tensor representation $Z^A$ and $Z^B$. Specifically, as shown in Figure 1 (a) $Z^A$

and $Z^B$ are two dimensional tensors, whereas in Figure 1 (b) $Z^A$ and $Z^B$ are three dimensional tensors due to an extra time dimension.

In order to address this difference, in this work we extend the application of Barlow–Twins loss for speech-to-text SSL modeling, in particular for speech (and more generally time varying) representation learning. As demonstrated in (Zbontar et al., 2021), for non time varying signals, one computes the cross-correlation matrix between the embeddings of online and target networks, and then optimizes the model such that this cross-correlation matrix comes close to the identity. For learning speech embeddings, we appropriately generalize the original Barlow–Twins loss by computing it from different views of the embedding data, namely as *time unrolling* and *time merging* approaches as depicted in Figure 1 (b), as well as in the text below.

i) **Time Unrolling.** In this approach, we unroll the sequence on time axis and stack each frame together. Specifically, a batch of outputs with dimensions $[B, T, F]$ is transformed to $[B * T, F]$. This transformation is performed for the outputs of both the online and target models. A cross correlation matrix $C^U$ of size $F \times F$ is then created and encouraged to be close to the identity via a loss function $L_U$. This enables the model to enforce diversity in each sequence, such that all frames cannot be the same. In other words, the time unrolling loss on the cross correlation matrix based on the reshaped tensor $[B * T, F]$ promotes that the $i - th$ feature across all the $B$ samples and $T$ timesteps is different from the $j - th$ feature.

ii) **Time Merging.** We merge together the features of each sequence on the time axis, so that the output of each utterance becomes one long tensor of features. We then compute a transpose of this as the Barlow–Twins loss is computed on the second dimension. Specifically, a batch of outputs with dimensions $[B, T, F]$ is transformed to $[T * F, B]$. Finally, we create a $B \times B$ correlation matrix $C^M$ and feed it to the loss function $L_M$. Through this loss we want the model to learn inter-sequence diversity such that the $i - th$ sample is different from the $j - th$ sample within the batch. This ensures that all sequences are not the same and thus SSL training does not collapse.

Formally, the loss function ($L$) can be defined as follows:

$$L_U \leftarrow \sum_i \frac{(1 - C_{ii}^U)^2}{N_U} + \sum_i \sum_{i \neq j} \frac{2(C_{ij}^U)^2}{N_U(N_U - 1)}, \text{ where } C^U \leftarrow \text{CrossCorrelation}(Z_U^A, Z_U^B)$$

$$L_M \leftarrow \sum_i \frac{(1 - C_{ii}^M)^2}{N_M} + \sum_i \sum_{i \neq j} \frac{2(C_{ij}^M)^2}{N_M(N_M - 1)}, \text{ where } C^M \leftarrow \text{CrossCorrelation}(Z_M^A, Z_M^B)$$

$$\text{loss} \leftarrow \begin{cases} w_U \cdot L_U + w_M \cdot L_M & \Leftarrow \text{static} \\ \frac{L_U}{\text{sg}(L_U)} + \frac{L_M}{\text{sg}(L_M)} & \Leftarrow \text{dynamic} \end{cases}$$

Here, $N_U$ and $N_M$ denote the first dimension of the square matrices $C^U$ and $C^M$ respectively. In practice we observe that the unrolling (U) loss $L_U$ and the merging (M) loss $L_M$ often have different scales, and for this reason we explore combining these loss functions statically or dynamically into an overall loss $L$. More specifically, for static scaling we simply multiply each loss function with a hyper-parameter weight $w_U$ for $L_U$ and $w_M$ for $L_M$ on which we conduct hyper-parameter optimisations (HPO). For dynamic scaling, we use the stop gradient (sg) operation to ensure each partial loss is divided by stop gradient applied to itself. Given that a scaling of the loss propagates to the size of the gradient update, this leads to overall better gradient update behavior without the need for HPO on the weight of each loss.

## 3 EVALUATION SETTINGS

### 3.1 ONLINE AND TARGET MODELS

We create the online and target networks via surgery on the Wav2Vec2 Base architecture (Baevski et al., 2020). This is accomplished by taking the feature extractor and the encoder components of Wav2Vec2 as well as additional augment and projection layers, which we organize in a sequential manner from feature extractor, augment layer, encoder and projection layer as described below.

**Feature Extractor.** The feature extractor consists of seven convolution blocks. Each block has a temporal convolution layer followed by layer normalization and a GELU activation function (Hendrycks

& Gimpel, 2016). The number of channels, strides and kernel widths are the same as in the Wav2Vec2 architecture (Baevski et al., 2020). This results in the feature extractor's hop length of $20ms$ and the receptive field of $25ms$ of audio.

**Augmentation in Latent Space.** An augmentation block is used to generate and apply masks on the feature extractor outputs. We use a masking function similar to SpecAugment (Park et al., 2019) and mask time-steps during training which avoid model collapse. We use the probability of $0.05$ and mask length of 10 for the target model, and we use $0.1$ and 20 for the online model. We use more masking for online network to bring highly distorted view close to its corresponding less distorted view, which is inspired from the local/global crop strategy presented in (Caron et al., 2021) and weak/strong augmentation strategy used in (Sohn et al., 2020). Note that, in order to avoids boundary effects, we apply augmentation only to the outputs of the feature extractor that mimics FFT style transformation with a receptive field of 25ms of audio and a 20ms hop length.

**Encoder.** We use the same encoder architecture as Wav2Vec2 Base model which consists of 12 transformer blocks.

**Projection.** We project the output to a lower dimensional latent space. Specifically, we use a dense layer to transform the output of the encoder to a 29 dimensional space. This helps us to efficiently compute the loss on smaller output dimensions. Here, we selected 29 dimensional output as it is close to the dimension of final vocabulary (discussed later in 3.4).

## 3.2 DATASET

During the SSL phase, we pre-train the models using the LibriSpeech corpus (Panayotov et al., 2015) of 960 hours of audio (LS-960) without the transcriptions. We crop the audio utterances into 5 seconds of speech and batch them together. We do not perform any pre-processing since the feature extractor directly processes the raw audio data.

When fine-tuning the models, we consider 3 labeled data settings:

i) LS-960: 960 hours of LibriSpeech data with transcriptions (Panayotov et al., 2015),

ii) LS-100: the train-clean-100 subset of LibriSpeech comprising 100 hours of labeled data,

iii) LL-10: Libri-light limited resource training subset of 10 hours labeled dataset, which was originally extracted from LibriSpeech (Kahn et al., 2020).

We evaluate models on the standard LibriSpeech dev-clean/other and test-clean/other sets.

## 3.3 PRE-TRAINING

For pre-training the representation learning model, that includes the feature extractor and encoder networks, we consider two settings as described below.

i) *Non-Contrastive*: this setting is based on our proposed non-contrastive SSL method for speech representation learning as described in Section 2.2.

ii) *Sequentially Combined*: in this setting, we consider pre-training the model with the *Wav2Vec2* approach followed by our proposed *Non-Contrastive* setting. Specifically, we start with pre-trained weights from Wav2Vec2 Base model (as described for *Wav2Vec2* pre-training setting) and further pre-train with non-contrastive SSL method (i.e., *Non-Contrastive* setting). Note that there could be multiple ways to combine the contrastive and non-contrastive training such as non-contrastive followed by contrastive, vice versa, or in an iterative sequence with each contrastive or non-contrastive being applied a fixed number of steps or perhaps with a scheduler to switch between the two approaches in a more elaborate fashion. However, we leave these for future exploration as they are outside of the scope of the present work.

**Optimization.** We use the ADAM optimizer (Kingma & Ba, 2015) with a learning rate of $10^{-5}$. The learning rate was selected between $10^{-5}$ and $5 \cdot 10^{-5}$ after training the models for $10K$ steps. We use exponential-decay with decay rate of $0.99$ as a learning rate scheduler. We train the Non-Contrastive and Sequentially Combined models for $400K$ and $250K$ steps respectively. For both of these methods we use the batch size corresponding to 16 minutes of audio (details are described in Table 4). We observed that the convergence with dynamic scaling of the loss (discussed in 2.2) was better than the static option, which motivated us to use it throughout our experiments. We note however that it is possible that the static option could work with other values of $w_M$ and $w_U$ than the

Table 1: Model performance in terms of WER on the Librispeech dev/test sets when fine-tuned on LS-960. Our *Non-Contrastive* and *Sequentially Combined* SSL methods achieve competitive performance compared with SOTA *Data2Vec2* model. Note WER results marked with '*' were computed by us based on the fine-tuned checkpoints and code from *Data2Vec2* since that paper did not contain WER results for those splits.

| Model | LM | dev | | test | |
|---|---|---|---|---|---|
| | | clean | other | clean | other |
| *Wav2Vec2* (Baevski et al., 2020) | None | 3.2 | 8.9 | 3.4 | 8.5 |
| *Data2Vec2 (Baevski et al., 2022a)* | None | 2.9* | 7.7* | 3.0* | 8.0 |
| *Non-Contrastive* | None | 3.1 | 8.8 | 3.3 | 8.7 |
| *Sequentially Combined* | None | 3.1 | 8.6 | 3.3 | 8.7 |
| *Wav2Vec2* (Baevski et al., 2020) | 4-gram | 2.0 | 5.9 | 2.6 | 6.1 |
| *Data2Vec2 (Baevski et al., 2022a)* | 4-gram | 2.0* | 5.6* | 2.5* | 5.2 |
| *Non-Contrastive* | 4-gram | 2.0 | 6.1 | 2.7 | 6.5 |
| *Sequentially Combined* | 4-gram | 2.0 | 6.1 | 2.7 | 6.4 |

ones we tried. Specifically we experimented with nine combinations with $w_M$ and $w_U$ taking values in [0.1, 0.5, 1.0], so a more extensive HPO could perhaps improve on the dynamic scaling.

## 3.4 FINE-TUNING

For the fine-tuning phase, we take the target network without the projection layer and add a randomly initialized linear projection on top. This layer's outputs have the same dimensionality as used in Wav2Vec2 Baevski et al. (2020). Moreover, we use the Wav2Vec2 fine-tuning recipes (namely Libri-10, Libri-100 and Libri-960) for training our models for downstream ASR task. This also ensures that we fine-tune the models for for LL-10, LS-100 and LS-960 with exact same amount of data and other settings.

## 3.5 DECODING AND LANGUAGE MODEL

For decoding, we use a CTC beam search decoder with and without language model (LM) to evaluate our models. A 4-gram LM trained on the LibriSpeech corpus is considered for decoding with the LM. We use beam width of 500 to measure the model performance with and without LM. We tune the weights of the LM for the range of $[0, 5]$ with an interval of $0.01$ and a word insertion penalty for the range of $[-5, 5]$ with an interval of $0.01$ on dev-other set.

## 3.6 METRICS

For ASR quality we consider the word error rate (WER) evaluation across various datasets, and for resource efficiency metrics we consider GPU hours and wall clock hours, as well as factors that influence these such as number of GPUs and batch size.

## 4 RESULTS

### 4.1 QUALITY COMPARISON OF NON-CONTRASTIVE AND MASK BASED CONTRASTIVE SSL

To examine the potential of our non-masking non-contrastive BT method for learning effective speech embeddings, we first compare its performance, for the downstream ASR task, with the SOTA masking based contrastive Wav2Vec2 (Baevski et al., 2020) as well as non-contrastive Data2Vec Baevski et al. (2022b) and Data2Vec2 Baevski et al. (2022a) methods.

We perform this evaluation by fine-tuning the pre-trained models in two settings. Firstly, with a *high-resource setting* where large quantities of labeled speech are available. This allows to compare the effectiveness of speech embeddings learned with the aforementioned SSL methods. Secondly, with a *low-resource setting* where the amount of labeled data is limited. This helps us compare the usefulness of speech embeddings, learned with the methods, for improving low resource settings.

Table 2: Model performance in terms of WER on the Librispeech dev/test sets when fine-tuned on LS-100 and LL-10. Our *Non-Contrastive* SSL method achieves significant improvement in WER as compared to the SOTA *Data2Vec2 Wav2Vec2* approaches. We highlight the top results for each LibriSpeech split with red color. Note that the non-contrastive and sequentially combined 4-gram LM results were not derived from extensive HPO on LM hyper-parameters as in other approaches. Note WER results marked with '†' were computed on fine-tuned models trained by us based on the pre-trained checkpoints and code from *Data2Vec2* since that paper did not contain WER results for those splits.

| Model | Fine-tuning Data | LM | dev | | test | |
|---|---|---|---|---|---|---|
| | | | clean | other | clean | other |
| *Wav2Vec2* (Baevski et al., 2020) | 10 | None | 10.9 | 17.4 | 11.1 | 17.6 |
| Data2Vec2 (Baevski et al., 2022a) | 10 | None | 7.3† | 12.3† | 7.4† | 12.7† |
| *Non-Contrastive* | 10 | None | 7.0 | 14.3 | 7.2 | 14.4 |
| *Sequentially Combined* | 10 | None | 6.1 | 12.7 | 6.4 | 12.4 |
| *Wav2Vec2* (Baevski et al., 2020) | 10 | 4-gram | 3.8 | 9.1 | 4.3 | 9.5 |
| Data2Vec2 (Baevski et al., 2022a) | 10 | 4-gram | 3.6† | 7.7† | 4.1† | 7.6 |
| *Non-Contrastive* | 10 | 4-gram | 3.3 | 8.8 | 3.8 | 9.3 |
| *Sequentially Combined* | 10 | 4-gram | 2.8 | 7.8 | 3.4 | 8.0 |
| *Wav2Vec2* (Baevski et al., 2020) | 100 | None | 6.1 | 13.5 | 6.1 | 13.3 |
| Data2Vec2 (Baevski et al., 2022a) | 100 | None | 4.1† | 9.0† | 4.2† | 9.3† |
| *Non-Contrastive* | 100 | None | 4.5 | 10.9 | 4.6 | 10.9 |
| *Sequentially Combined* | 100 | None | 4.1 | 10.3 | 4.2 | 9.6 |
| *Wav2Vec2* (Baevski et al., 2020) | 100 | 4-gram | 2.7 | 7.9 | 3.4 | 8.0 |
| Data2Vec2 (Baevski et al., 2022a) | 100 | 4-gram | 2.3† | 6.4† | 2.5† | 6.4 |
| *Non-Contrastive* | 100 | 4-gram | 2.3 | 7.0 | 2.9 | 7.4 |
| *Sequentially Combined* | 100 | 4-gram | 2.2 | 6.5 | 2.7 | 6.7 |

**Evaluation with High-Resource Labeled Data.**  We fine-tune the embedding models, pre-trained using *Non-Contrastive* and *Sequentially Combined* methods (discussed in Section 3.3), with LS-960 dataset. Note that even though the *Sequentially Combined* approach is computationally inefficient, we investigate it to examine the potential of sequentially combined pre-training when resources are not a constraint or when a pre-trained model is available and we would like to further improve it.

Table 1 presents the ASR model performance in terms of WER. The results for these evaluations show *our proposed Non-Contrastive SSL method achieves competitive performance compared to the SOTA approaches for speech representation learning on LibriSpeech dataset.*

At the same time, our results show the *Sequentially Combined SSL approach significantly boosts the performance of downstream ASR task and in addition achieves up to 0.2% absolute WER improvement on dev/test-other splits.* Conclusively, if the priority is to decrease WER rather than training time, Sequentially Combined SSL method is observed to yield the better results.

**Evaluation with Low-Resource Labeled Data.**  To evaluate the performance in low resource settings, we considered fine-tuning our pre-trained models with LS-100 and LL-10 datasets (discussed in Section 3.2.

Table 2 presents the ASR model performance on low resource settings in terms of WER on LibriSpeech dev and test splits. Our results demonstrate that even on the low resource settings, the speech embeddings learned on unlabeled data with our non-contrastive SSL method are significantly more effective than the ones learned with Wav2Vec2 and Data2Vec2 SSL methods. Specifically, *we observe our model yields new SOTA for multiple splits of LS-100 and LL-10 datasets.* Note that we

Table 3: Computation time for pre-training with different SSL approaches. Compared to *Data2Vec2*, our *Non-Contrastive* approach requires $1.3\times$ less GPU hours. With a fraction of extra computation time, we can further boost the performance of *Wav2Vec2* using *Sequentially Combined* method.

| Model | GPU Type | GPU Hours | No. of GPUs | Wall Clock Hours |
|---|---|---|---|---|
| *Data2Vec2* | A100 | 1376 | 32 | 43 |
| *Non-Contrastive* | V100 | 1064 | 8 | 133 |
| *Sequentially Combined* | V100 | 3040 (2376+664) | 8 | 380 (297+83) |

Table 4: Batch size requirements for pre-training with different SSL approaches. Our results show that *Non-Contrastive* SSL method requires smaller batch sizes as compared to *Data2Vec2* and *Wav2Vec2* approaches. For non-contrastive method we use the batch size per GPU of 120 secs by leveraging the gradient accumulation of 3 steps each with batch size of 40 secs.

| Model | GPU Type | No. of GPUs | Batch size/GPU (secs) | Batch size (secs) |
|---|---|---|---|---|
| *Wav2Vec2* | V100 | 8 | 84 ($\times 8$) | 5376 |
| *Data2Vec2* | V100 | 8 | 32 ($\times 4$) | 1020 |
| *Non-Contrastive* | V100 | 8 | 40 ($\times 3$) | 960 |
| *Wav2Vec2* | A100 | 32 | 174 | 5580 |
| *Data2Vec2* | A100 | 32 | 32 | 1020 |
| *Non-Contrastive* | A100 | 32 | 30 | 960 |

Table 5: Training cost (in terms of GPU hours) for Non-Contrastive SSL method with different batch sizes and sequence lengths. Results are computed when training on 8 V100 GPUs with gradient accumulation of 3 steps. We observe that increasing the batch size results in a higher computation time than increasing the sequence lengths. Similarly, the impact on WER is reported in the Appendix.

| Batch Size | GPU Hours | | |
|---|---|---|---|
| | Seq Len 3s | Seq Len 5s | Seq Len 7s |
| **24** | 704 | 1064 | 1483 |
| **48** | 1216 | OOM | OOM |
| **72** | 1696 | OOM | OOM |

did not perform extensive HPO on LM hyper-parameters for the evaluation with LS-100 and LL-10 as in Wav2Vec2 and Data2Vec2 approaches. Furthermore, we also observe that our *Sequentially Combined* method consistently improves upon our *Non-Contrastive* approach.

## 4.2 COMPUTATIONAL AND RESOURCE EFFICIENCY

There has been great progress in the improvement of ASR systems which are founded through modeling advancements (e.g., Conformer (Gulati et al., 2020)), pre-training and learning methodologies (e.g.,Wav2Vec2 (Baevski et al., 2020)), data augmentations (e.g., SpecAugment (Park et al., 2019)), and so on. Nevertheless, training time and the requirements for a huge number of resources remain a significant barrier to scale up these improvements for a wider number of practitioners, supported languages, latency requirements and deployment scenarios.

Due to this reason, in this section we investigate the computational and resource efficiency of non-contrastive versus contrastive SSL approaches for speech representation learning. Table 3 shows the quantification of the GPU hours required for pre-training with these SSL methods, and Table 4 presents the batch size requirements which is a key enabler of allowing for better embedding models at the cost of increased memory requirement.

As shown in Table 3, our results demonstrate that *indeed in SSL for ASR, Data2Vec2 training time can be decreased via our non-contrastive SSL by as much as* $1.3\times$ *(1376 / 1064).* Additionally, inline with the previous study (Zbontar et al., 2021), where contrastive approaches have been found to require much larger batch sizes than non-contrastive approaches. Our experimental setup, as described in

Table 4, also highlights *smaller batch size requirements for our non-contrastive approach (960 secs) as opposed to Wav2Vec2 (5376 secs) and Data2Vec2 (1020 secs) approaches, allowing for training on memory constrained GPUs*. Note that in principle it is possible to train Data2Vec2 with smaller batch size (to be comparable with our approach), but it could impact the WER of the fine-tuned model. Specifically, when training Data2Vec2 ourselves, we observed that by reducing the batch size by $4\times$, the WER could increase by up to 10% relatively.

Finally, it is worth noting that if the priority is to decrease WER rather than training time or GPU resources, lowest WER can be achieved by sequentially combined contrastive followed by non-contrastive training. Although, we have already discussed WER improvements from this approach in the previous subsection, we would like to highlight that the increase in computation time and batch size cost is only of a small fraction as can be seen in Tables 3 and 4.

### 4.3 Impact of Batch Size and Sequence Length on Computation Time

In this subsection, we perform an ablation study to understand the impact of the batch size and sequence length on training time. In particular, having as a reference the setting considered in Wav2Vec2, namely batch size 6 with approx. 15s in each sample resulting in a lower amount of audio per batch of 1.4min. We expand our measurements from a single setting of batch size 24 with 5s in each sample with yields 2min of audio per batch to different configurations of batch sizes 24, 48 and 72 as well as sequence lengths of 3s, 5s and 7s. We limit the exploration to maximum batch size of 72 and sequence length of 7s as allowed by our infrastructure (with NVIDIA V100 GPUs) without increasing the gradient accumulation steps (fixed at three as in other experiments).

As shown in Table 5, we observe that increasing the batch size results in a higher computation time than increasing the sequence lengths. This may be due to GPU contention at higher batch sizes, or that depending on the configuration of batch size and sequence length different kernels for the different operations are being selected by the backend. Moreover, given that the non-contrastive approach is based on a siamese network format, with exponential moving average, there are twice the weights in memory as well as twice the number of forward passes which limits how much we can increase the batch size and the sequence lengths. Therefore, even with three step gradient accumulation (as used in all other experiments in the paper) we can see in the table that the more demanding settings result in out of memory (OOM) on V100 GPUs having 32 GB memory.

### 5 Conclusion

This work made a number of contributions towards resource efficient speech representation learning for ASR. First, we investigated the potential of replacing established masking based contrastive and non-contrastive SSL approaches (e.g., Wav2Vec2 (Baevski et al., 2020) and Data2Vec2 (Baevski et al., 2022a)) with a non-masking non-contrastive SSL method to reduce training time and GPU resources needed for training ASR systems, without compromising, actually improving, model quality. Second, we achieved this goal by extending existing image based non-contrastive Barlow–Twins (Zbontar et al., 2021) method to speech representation modeling and more generally to arbitrary time varying data. Numerical results demonstrated a training speed up of $2.23\times$ relative to Wav2Vec2, almost doubling the speed improvement achieved by Data2Vec2, which together with lower VRAM requirements allows our method to run on more affordable compute infrastructure as well as improved model performance (i.e., 44%, 32% and 3% relative improvements on LibriSpeech 10, 100 and 960 hours datasets, respectively). Finally, our results demonstrated the benefits of sequentially combining masking contrastive Wav2Vec2 and non-masking non-contrastive BT methods to further boost ASR performance.

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
