# OpenReview forum: "Resource Efficient Self-Supervised Learning for Speech Embeddings"
_ICLR.cc/2024/Conference — Submitted to ICLR 2024_

### Official Review · Reviewer_3Nqf · 2023-10-21

**Soundness:** 2 fair
**Presentation:** 2 fair
**Contribution:** 2 fair
**Rating:** 3
**Confidence:** 4

**Summary:**

This paper proposes a non-contrastive self-supervised learning (SSL) method for speech models. The core idea of this method is to leverage the Barlow-Twins (BT) training technique, which tracks the EMA of the model and compares it with the original to compute loss. Specifically, the method encourages the correlation matrix to become identity, both in a batch-wise and time(frame)-wise manner. Because additional time-axis presents in speech data, the authors show two ways to incorporate time information (time unrolling and time merging). By combining these losses with latent-level augmentation (like SpecAug) and hyper-parameter optimization, the proposed method achieves comparable speech recognition performance to previous models such as Wav2Vec2 and Data2Vec2 while utilizing fewer GPU resources.

**Strengths:**

* This paper appears to be the first adaptation of a BT-like approach for speech SSL domain. The method naturally integrates the time dimension, making the method more tailored to speech.
* It is good to see the research that addresses the resource-intensive nature of SSL training. Especially, the suggestion of training with shorter sequence length is a novel contribution not much explored in previous works.

**Weaknesses:**

* Despite the title of this paper mentioning “speech embeddings”, experiments only evaluate the learned representations in the context of the ASR task. A broader evaluation including common SSL benchmarks such as SUPERB[1] would provide a more comprehensive understanding.
* Related works on SSL do not include recent SSL papers such as WavLM[2] or BEST-RQ[3].
* I am not quite sure that speech SSL methods are either contrastive or non-contrastive. For example, HuBERT is more like a BERT-style Masked Language Modeling (MLM) approach rather than a contrastive one. This line of work includes w2v-BERT, WavLM, and BEST-RQ. As HuBERT / WavLM are gaining popularity, I think a comparison with these methods would be beneficial.
* The authors discuss static and dynamic scaling techniques for balancing the loss; however, there are no corresponding experimental results.
* (minor) The reference style is inconsistent across different sections of the paper, which makes it difficult to read. Please consider unifying the style throughout the paper.

**References**

[1] SUPERB: Speech Processing Universal PERformance Benchmark\
[2] WavLM: Large-Scale Self-Supervised Pre-training for Full Stack Speech Processing\
[3] BEST-RQ: Self-Supervised Learning with Random-Projection Quantizer for Speech Recognition

**Questions:**

* The performance gap of ‘non-contrastive’ vs. ‘sequentially combined’ is not small for low-data scenarios. This raises the question of whether the non-contrastive approach sufficiently provides information for speech recognition. Is the performance gap caused by contrastive learning’s ability, or, is it a by-product of longer training? I’d like to hear your thoughts on this.
* Regarding the numbers in Table 3, for sequentially combined cases, where do 3040 (2376+664) and 380 (297+83) come from? It does not seem that there are clear explanations for these numbers.
* The authors mention the Appendix in Table 5, but I cannot find the Appendix.

---

> ### Author Response · Authors · 2023-11-23
>
> > Despite the title of this paper mentioning “speech embeddings”, experiments only evaluate the learned representations in the context of the ASR task. A broader evaluation including common SSL benchmarks such as SUPERB[1] would provide a more comprehensive understanding.
>
> We have added some results and are planning to add more if time permits.
>
> | Model | SLU | KS |
> | -------- | ------- | ------- |
> | Data2vec-2 | 73.3% | 98.5% |
> | Wav2vec-2 | 73.1% | 98.0% |
> | Ours | 73.4% | 98.0% |
>
>
>
> > Related works on SSL do not include recent SSL papers such as WavLM[2] or BEST-RQ[3].
>
> Thanks for the suggestion. We will extend the related work and add a proper discussion about these works in the related work.
>
> > I am not quite sure that speech SSL methods are either contrastive or non-contrastive. For example, HuBERT is more like a BERT-style Masked Language Modeling (MLM) approach rather than a contrastive one. This line of work includes w2v-BERT, WavLM, and BEST-RQ. As HuBERT / WavLM are gaining popularity, I think a comparison with these methods would be beneficial.
>
> We agree with the reviewer that comparison with other masking based SSL approaches would benefit the paper. We did not add the other baselines since Data2Vec-2 showed SOTA compared to all others. However, we will add the suggested SSL approaches to the paper and compare them with our approach.
>
> > The authors discuss static and dynamic scaling techniques for balancing the loss; however, there are no corresponding experimental results.
>
> We were not able to perform ablation on those two loss scalings since we couldn’t converge the model with static scaling. We will clarify this in the paper.
>
> > (minor) The reference style is inconsistent across different sections of the paper, which makes it difficult to read. Please consider unifying the style throughout the paper.
>
> Thanks for pointing this out. We will fix this issue.
>
> > The performance gap of ‘non-contrastive’ vs. ‘sequentially combined’ is not small for low-data scenarios. This raises the question of whether the non-contrastive approach sufficiently provides information for speech recognition. Is the performance gap caused by contrastive learning’s ability, or, is it a by-product of longer training? I’d like to hear your thoughts on this.
>
> We observed that during the pre-training the models do not converge further after around 350k steps. Therefore, we believe that the performance gain comes with the combination of contrastive and non-contrastive learning.
>
> > Regarding the numbers in Table 3, for sequentially combined cases, where do 3040 (2376+664) and 380 (297+83) come from? It does not seem that there are clear explanations for these numbers.
>
> 3040 = 2376 (i.e., GPU hours for Wav2Vec-2) + 664 (i.e., GPU hours with our approach).
> 380 = 297 (wall clock time for Wav2Vec-2) + 83 (wall clock time for our approach)
> We will clarify this in the paper.
>
> > The authors mention the Appendix in Table 5, but I cannot find the Appendix.
>
> That’s our fault. We will fix this in the revised version of the paper.

---

### Official Review · Reviewer_iuxy · 2023-10-31

**Soundness:** 3 good
**Presentation:** 3 good
**Contribution:** 3 good
**Rating:** 6
**Confidence:** 4

**Summary:**

The paper presents a non-contrastive learning approach to self-supervised learning from speech. More specifically, the proposed method extends the Barlow-Twins methodology so the loss is defined over sequential data. The method also requires fewer resources, while providing competitive performances, especially in low-resource languages.

**Strengths:**

- The proposed time unrolling and time merging methods appear to be an adequate extension of the Barlow-Twins method, which was originally proposed for non-sequential data.

- The proposed method shows consistent performance; it either competes with the sota performance or improves it.

- The overview of the proposed method is summarized well in Fig 1.

**Weaknesses:**

- It's not clear which of the two terms is contributing more to the performance of the model, in the proposed loss function L_U and L_M.

- In general, the paper doesn't provide an in-depth justification for the claims, and relegate the explanation to the reference, such as the implication and importance of gradient stopping.

- It seems that the proposed method could save some GPU time, but not too significantly (1.3X less).

- The presentation of the loss function is somewhat abrupt, lacking explanation.

- Literature review could be more organized.

**Questions:**

- One of the main claims is that the proposed method provides a new SOTA result on the low-resource labeled data, which is good. However, it's not clear why the proposed method cannot compete with the Data2Vec2 method in the high-resource labeled data experiments.

---

> ### Author Response · Authors · 2023-11-23
>
> > It's not clear which of the two terms is contributing more to the performance of the model, in the proposed loss function L_U and L_M. In general, the paper doesn't provide an in-depth justification for the claims, and relegate the explanation to the reference, such as the implication and importance of gradient stopping.
>
> We were not able to perform ablation on those two loss scalings since we couldn’t converge the model with static scaling. Therefore, we could not even show the individual contribution of the two losses. We will clarify this in the paper.
>
> > It seems that the proposed method could save some GPU time, but not too significantly (1.3X less).
>
> We agree with the reviewer that mentioning this absolute improvement in that way may look less exciting or impactful. However, we also wish to highlight one important thing: training SSL models is notoriously expensive, and a 1.3X improvement in speed can very quickly translate into days of training and cost savings, especially from a multi-GPU training perspective. We also know well that deployed or usable SSL models are not the outcome of a single training run, meaning that these needed resources must be multiplied by the number of trials. This is a 1.3X reduction in time, energy consumption and cost. From this regard, any improvement is something positive for the community. The refinement of today’s models comes from the progressive incremental improvements of training efficiency [1] – and we believe that a 1.3X improvement already is a significant step for anyone interested in training a SSL model with a limited GPU budget.
>
> [1] Vyas, A., Hsu, W. N., Auli, M., & Baevski, A. (2022). On-demand compute reduction with stochastic wav2vec 2.0. arXiv preprint arXiv:2204.11934.
>
> > The presentation of the loss function is somewhat abrupt, lacking explanation.
> > Literature review could be more organized.
>
> Thanks for the suggestion. We will address these comments in the revised version of the paper.
>
> >One of the main claims is that the proposed method provides a new SOTA result on the low-resource labeled data, which is good. However, it's not clear why the proposed method cannot compete with the Data2Vec2 method in the high-resource labeled data experiments.
>
> We believe that Data2Vec-2 was well optimised to achieve SOTA WER results, which require a very thorough hyper-parameter optimization (HPO). However, we do not perform rigorous HPO due to limited computational resources. Still we show that our approach marginally higher WER for LibriSpeech splits, except for dev-clean where all methods achieve the same WER (i.e., 2%).

---

### Official Review · Reviewer_QXLT · 2023-11-01

**Soundness:** 3 good
**Presentation:** 3 good
**Contribution:** 2 fair
**Rating:** 6
**Confidence:** 3

**Summary:**

- self-supervised learning (SSL) approaches have helped to improve speech model performance
- these techniques have included both contrastive and non-contrastive methods
- non-contrastive methods (like Data2Vec2 vs Wav2Vec2) have improved performance and reduced training time, but still suffers from significant GPU resource constraints
- this work introduces a non-contrastive approach which is an extension of the Barlow-Twins methodology to reduce training time and resource requirements for self-supervised model training
- to adapt speech for the Barlow-Twins method, they use time-unrolling ([B, T, F] -> [B * T, F]) and time-merging ([B, T, F] -> [T * F, B]) approaches when computing two different cross-correlation terms for the losses
- the primary comparisons are done with wav2vec2, data2vec2, non-contrastive (their approach), and sequentially combined (wav2vec2 training then non-contrastive)
- these comparisons are done when fine-tuning on LibriSpeech 960h, train-clean-100h, and LibriLight-10h
- 960h: the results are slightly better w/o LM but w/ LM they're competitive
- for the 10h and 100h settings, the result trends broadly follow:
         - non-contrastive consistently outperforms wav2vec2
         - sequentially combined outperforms non-contrastive
         - but sequentially combined and data2vec2 are more competitive with each other
- the main benefits come from requiring less resources by being able to use smaller batch sizes (reducing training time) as well as fewer GPUs

**Strengths:**

Straightforward motivation, modification/adaptation of an existing idea, and execution. Primarily isolating changes to the loss, while keeping architectural changes minimal.

**Weaknesses:**

Despite the similarities to Wav2Vec2 and Data2Vec2, it would be nice to include more non-contrastive comparisons (especially since the application focus is on speech), these would ideally include at least one of HuBERT and WavLM. Since the takeaway here seems to be about reducing resource requirements while maintaining high-quality performance, comparison with these popular approaches both in terms of training resources required and inclusion in the WERs table would be helpful.

Also, in the abstract, SSLs are mentioned as being great for a variety of tasks. Seeing the performance of this non-contrastive approach on not only ASR but other speech tasks as well could help to distinguish it from Data2Vec2 (since it often needs to be sequentially combined with wav2vec2 to match Data2Vec2 performance on the "other" partition of dev or test). These SSLs are useful in a variety of cases, so an idea of the general performance hit (in service of resource savings) on these other tasks would be helpful.

**Questions:**

Resource and performance comparisons with HuBERT and WavLM would be helpful (for more non-contrastive comparisons).

It would be nice to see the trade-off between training time (or # of GPUs) and performance between these models (i.e. if you speed up the training recipe of the contrastive vs non-contrastive approaches does the performance degrade similarly in cases where you have even more significant resource constraints than those given).

---

> ### Author Response · Authors · 2023-11-23
>
> > Despite the similarities to Wav2Vec2 and Data2Vec2, it would be nice to include more non-contrastive comparisons (especially since the application focus is on speech), these would ideally include at least one of HuBERT and WavLM. Since the takeaway here seems to be about reducing resource requirements while maintaining high-quality performance, comparison with these popular approaches both in terms of training resources required and inclusion in the WERs table would be helpful.
>
> We agree with the reviewer that comparison with HuBERT and WavLM as baselines would benefit the paper. We did not add the other baselines since Data2Vec-2 showed SOTA compared to all others. However, we will add the suggested SSL approaches to the paper and compare them with our approach in table 1, 2, 3, and 4.
>
> > Also, in the abstract, SSLs are mentioned as being great for a variety of tasks. Seeing the performance of this non-contrastive approach on not only ASR but other speech tasks as well could help to distinguish it from Data2Vec2 (since it often needs to be sequentially combined with wav2vec2 to match Data2Vec2 performance on the "other" partition of dev or test). These SSLs are useful in a variety of cases, so an idea of the general performance hit (in service of resource savings) on these other tasks would be helpful.
>
> Following the reviewer's advice, we added two new tasks from the SUPERB benchmark to compare our model to wav2vec2 and data2vec2: Spoken Language Understanding with SLURP and Keyword Spotting with Google Speech command. Following the open-source recipe of these tasks on SpeechBrain we fine-tuned three different models to obtain the results reported on the following Table. SLURP results are reported with the introduced SLU-F1 score while Keyword Spotting is a simple accuracy. In both cases, higher is better. Finally, we would like to highlight that other accepted articles on SSL, such as BEST-RQ, are also only evaluated on ASR.
>
> | Model | SLU | KS |
> | -------- | ------- | ------- |
> | Data2vec-2 | 73.3% | 98.5% |
> | Wav2vec-2 | 73.1% | 98.0% |
> | Ours | 73.4% | 98.0% |

---

### Official Review · Reviewer_CDH9 · 2023-11-05

**Soundness:** 2 fair
**Presentation:** 3 good
**Contribution:** 3 good
**Rating:** 5
**Confidence:** 4

**Summary:**

While wav2vec-style contrastive learning has shown to be very successful for ASR, it requires a lot of resources and time for training. In the vision domain, Barlow Twins, a solution that naturally avoids collapse, has shown to be able to achieve better (or competitive) performance compared with contrastive learning (e.g., SimCLR) while using much smaller batch size. However, Barlow Twins style training is under-explored in the ASR/audio domain.

The authors explored using BT to speech representation learning and achieved competitive performance compared with wav2vec2; The authors further combined their approach with wav2vec2 to further boost the performance. The authors claim the proposed methods can reduce training time, GPU usage and improve convergence.

**Strengths:**

This is, if not the first, among the early explorations that applies Barlow-Twins methodology to learn representation for ASR; The adoption of BT into sequential representation learning is not trivial. Previously, BT was mostly used to learn a global representation of a sequence.



— The authors show that the proposed methods are more resource efficient and achieve comparative performance. a) It improves convergence, b) it reduces training time, c) it significantly reduces GPU training times, d) it requires smaller batch size thus reducing memory requirements.



— The authors also found that combining the proposed method with a wav2vec-style contrastive learning approach is helpful.

**Weaknesses:**

The proposed approach, though has some computational benefits when compared to Data2vec2, it achieves clearly worse performance compared to Data2vec2.

The authors try to combine wav2vec2 pre-training and the proposed method, which can significantly improve the performance, but the performance is still worse than Data2vec2. What's more, after combining with wav2vec2 pre-training, the computational resources needed would increase drastly.





— Regarding Time Unrolling and time merging losses: To calculate both the F by F and B by B correlation matrix, the calculation can become a burden when sequence length T is big. In this work, the authors propose to crop the audio into 5-seconds. However, these limitations could affect learning in a large context.



The authors do not test their model on tasks other than ASR.

**Questions:**

See Weaknesses section

---

> ### Author Response · Authors · 2023-11-23
>
> > The proposed approach, though has some computational benefits when compared to Data2vec2, it achieves clearly worse performance compared to Data2vec2. The authors try to combine wav2vec2 pre-training and the proposed method, which can significantly improve the performance, but the performance is still worse than Data2vec2. What's more, after combining with wav2vec2 pre-training, the computational resources needed would increase drastly.
>
> We would like to highlight that training SSL models is notoriously expensive, and a 1.3X improvement in speed can very quickly translate into days of training and cost savings, especially from a multi-GPU training perspective. We also know well that deployed or usable SSL models are not the outcome of a single training run, meaning that these needed resources must be multiplied by the number of trials. This is a 1.3X reduction in time, energy consumption and cost, which is significant for a large-scale SSL training. From this regard, any improvement is something positive for the community. At the same time,  we do not claim SOTA for all LibriSpeech splits, but we do show our model achieves competitive results. Additionally, our model also achieves SOTA on low resource settings.
>
>
> > Regarding Time Unrolling and time merging losses: To calculate both the F by F and B by B correlation matrix, the calculation can become a burden when sequence length T is big. In this work, the authors propose to crop the audio into 5-seconds. However, these limitations could affect learning in a large context.
>
> We agree with the reviewer that large sequence lengths could add an overhead for loss computation. However, we do not observe any impact on WER when using 7 secs sequence length compared to 5 secs. In particular, we based this choice on a published article investigating the impact of variable sequence length for SSL pretraining: “Match to Win: Analysing Sequences Lengths for Efficient Self-supervised Learning in Speech and Audio” from SLT 2022.
>
>
> > The authors do not test their model on tasks other than ASR.
> Following the reviewer's advice, we added two new tasks from the SUPERB benchmark to compare our model to wav2vec2 and data2vec2: Spoken Language Understanding with SLURP and Keyword Spotting with Google Speech command. Following the open-source recipe of these tasks on SpeechBrain we fine-tuned three different models to obtain the results reported on the following Table. SLURP results are reported with the introduced SLU-F1 score while Keyword Spotting is a simple accuracy. In both cases, higher is better. Finally, we would like to highlight that other accepted articles on SSL, such as BEST-RQ, are also only evaluated on ASR.
>
> | Model | SLU | KS |
> | -------- | ------- | ------- |
> | Data2vec-2 | 73.3% | 98.5% |
> | Wav2vec-2 | 73.1% | 98.0% |
> | Ours | 73.4% | 98.0% |

---

### Meta-Review · Area_Chair_e9a8 · 2023-12-04

**Metareview:**

The authors propose to extend the Barlow-Twins solution, so that the loss can be defined over sequential data. In doing so, the proposed approach uses fewer resources, while providing competitive performances, especially in low-resource languages. It however seems that the advantages of the proposed technique are not always clear - perhaps because the ASR-specific settings as pointed out by one of the reviewers, and additional experimental evaluation might be needed.

**Justification For Why Not Higher Score:**

It is an interesting idea, but  some points need further analysis.

**Justification For Why Not Lower Score:**

N/A

---

### Decision · Program_Chairs · 2024-01-16

Reject